# Towards annual updating of forced warming to date and constrained climate projections

A. Ribes [1,10] ✉, O. Tessiot [1,2,10], P. M. Forster [3], N. P. Gillett [4], V. Masson-Delmotte [5], J. Rogelj [6,7], R. Vautard[8] & T. Walsh [9]

In the context of rapid human-caused climate change, regular updates of the state of knowledge of current and future climate are needed. New statistical methods using observational constraints underpinned estimates of present-day human-induced warming and projected future warming in the most recent IPCC report. As time goes by, and new updated observational records become available, how should estimates of the current and projected human-caused climate change be updated? Here, we use a perfect model framework and show that incorporating observations from every new year in observationally constrained projections improves their accuracy, without causing major year-to-year spurious variability on outcomes. The forced warming estimated for the current year also exhibits high enough stability to be considered as a robust indicator of the state of the climate system.

As current warming is approaching the lower, 1.5 °C limit of the Paris Agreement long-term temperature goal, there is a growing appetite to understand how climate change is unfolding and how fast it actually changes, in near real time. From this perspective, regular updates to the current state of the climate at global, regional or national scales are extremely useful. Several organisations carry out regular climate monitoring on a global scale, describing temperature variations across the globe on a yearly, monthly or even daily basis[1,2]. However, the observed temperature is the result of internal variability (up to a few tenths of a degree on the global average, and even more regionally, including interannual to multidecadal fluctuations) superimposed on top of the forced response[3,4]. A specific estimation of the forced component is required to characterise climate change to date, and to compare its current state with long-term global warming levels used in international climate negotiations[5–8]. Separation of human and natural forced drivers of climate change, as well as internal variability, is also useful for understanding recent observations of surface temperature change. It is also important to understand what are the implications of the forced response to date for future climate.

The 6th Assessment Report[9] (AR6) of Working Group I (WGI) of the Intergovernmental Panel on Climate Change (IPCC) used the average warming over the last observed 10 or 20 years to characterise the current state of the climate system[10]. In a context of rapid warming, this choice may not be optimal[7,11], as the estimated warming level lags behind the changes that have occurred to date. It also has policy and communication implications as the world is getting closer to the 1.5 °C global warming limit included in the Paris Agreement. A recent update[11] estimated both the human-induced warming over the last 10 years ( + 1.19 [1.0 to 1.4] °C over 2014–2023, hereafter the 10-year estimate), and 1.31 [1.1 to 1.7] °C for the year 2023 (hereafter the 1-year estimate). The difference between these two estimates suggests a (human-induced) warming of about 0.1 °C higher in 2023 than in the preceding decade, 2014–2023, a result fully aligned with the estimated decadal global warming rate of about 0.26 °C/decade over 2014-2023[11,12]. This difference raises the question: which of these two estimators is the most accurate at characterising the forced warming experienced at present?

[1]Météo France, CNRS, Univ. Toulouse, CNRM, Toulouse, France. [2]Direction de la Climatologie et des Services Climatiques, Météo-France, Toulouse, France. [3]Priestley Centre for Climate Futures, University of Leeds, Leeds, UK. [4]CCCma, Environment and Climate Change Canada, Victoria, BC, Canada. [5]LSCE (UMR 8212 CEA-CNRS-UVSQ), Institut Pierre Simon Laplace, Université Paris Saclay, Gif-sur-Yvette, France. [6]Centre for Environmental Policy and Grantham Institute – Climate Change and Environment, Imperial College London, London, UK. [7]Energy, Climate and Environment Program, International Institute for Applied Systems Analysis, Laxenburg, Austria. [8]Institut Pierre-Simon Laplace, CNRS, Université Paris-Saclay, Sorbonne Université, Paris, France. [9]Environmental Change Institute, University of Oxford, Oxford, UK. [10]These authors contributed equally: A. Ribes, O. Tessiot. ✉e-mail: aurelien.ribes@meteo.fr

Updating forced warming estimates is also applicable to projections of the future climate. Until AR6, climate projections for the 21st century and beyond typically relied on raw simulations from climate models, with no direct use of observations[13]. The AR6 saw a change of approach, with a narrowing of the large range of CMIP6 climate responses based on historical observations and independently assessed climate sensitivity ranges grounded in multiple lines of evidence[12,14]. Assessed global projections thus rely on observational constraints, i.e., the filtering of relevant climate responses on the basis of available observations[11]. Through this mechanism, constrained climate projections also become a function of recent observations. This naturally raises the possibility of regular updates of observationally-constrained projections, alongside other key indicators based on observations[11]. As a recent example, the very high global-mean surface temperature (GST) of the second half of 2023 and 2024 has raised fair questions about their long-term implications, and is already the subject of a large body of literature[15–20]: are such warm years consistent with the previously estimated forced warming trajectory? or should this trajectory be revised upwards? These questions regarding the current and future climate response are of high policy relevance.

As the IPCC reports are usually published in cycles of 5–7 years, and the AR7 WGI Reports would be published only in the last years of this decade, more regular updates are needed. These could help authors to trace recent developments, to contextualise the latest observations, and to assess remaining carbon budgets with increased accuracy. A recent initiative[11,21] provided annual updates of several key indicators of the state of the global climate system for the first time, covering both societal (e.g., greenhouse gas emissions) and physical aspects (e.g., updated estimates of global surface temperature, radiative forcings, total forced and human-induced warmings, Earth's energy imbalance). Updates for this set of indicators will be published annually and are presented in an accompanying dashboard to provide an up-to-date description of the state of the climate.

In order to support and extend this effort, we investigate two intertwined questions. First, is the estimated forced warming to date a robust indicator? Second, should long-term projections be constrained by the latest available observations? Here, we investigate the influence of year-to-year internal variability[4], both at the global and regional scales, to assess the strength and limitations of using these indicators in near real time.

Our analysis is based on an observational constraint method, called Kriging for Climate Change (KCC), that seamlessly estimates past, present and future forced warming in response to different emission scenarios[22–24] (see Fig. 1 and "Methods"). This method involves mainly a Kalman filter (or Kriging) of a range of climate model projections. Its implementation requires a careful estimation of the uncertainties related to climate models and observations (see Methods). This technique was one of the techniques assessed and applied in the attribution and projection chapters of the AR6[10,12] and in subsequent updates[11,21]. Nevertheless, key conclusions of this study are expected to also hold for other statistical methods used to constrain projections[25–28] or estimate attributable warming[7,29].

## Results
### Forced warming to date
Here, we focus on total forced warming in response to both anthropogenic and natural forcings. The total forced warming was very close to the human-induced warming over the last decade[10,11,21,22,29], and will remain so in the future except in the aftermath of major volcanic eruptions, so our conclusions on the stability of estimators would also apply to human-induced warming. By *warming to date*, we mean the forced warming for the last observed (i.e., preceding) year relative to preindustrial[7,11,21,22]. Considering the current year would also be possible and defensible, and would not substantially affect the key results of this study. We investigate how to best estimate the warming to date, and in particular, we compare the 10-year and 1-year estimates discussed above. These estimators are both derived from an attribution method, aimed at separating forced response and internal variability – they are not just the observed warming over 1-year or 10-year. Recent attribution methods do provide 1-year estimates, i.e., an estimate of the influence of a subset of external forcings for one specific year[7,11,21,22]. This calculation is distinct from the linear extrapolation used in the IPCC SR1.5[30] – the end point of a linear trend fitted to the observations over the most recent 15 years – but results in extremely similar values

## Global temperature constraint

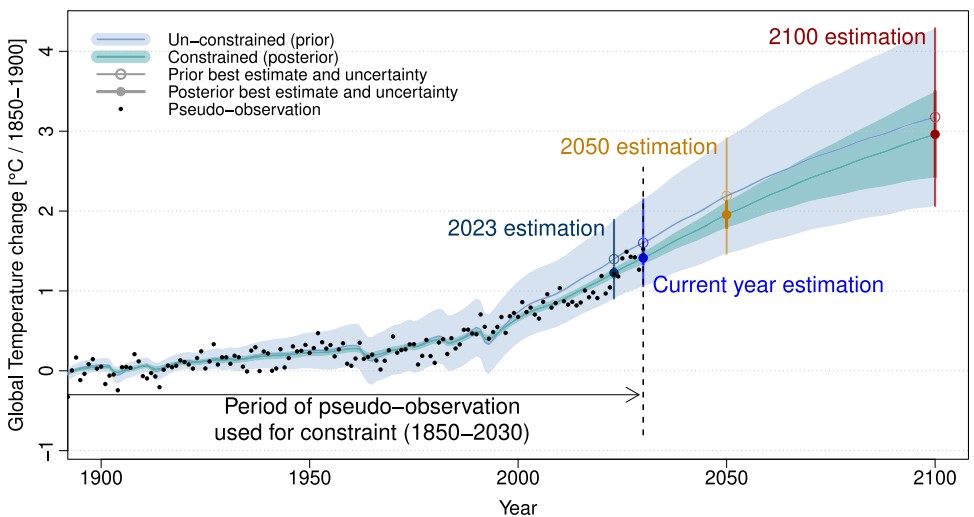

**Fig. 1 | Illustration of the observational constraint procedure.** This illustration is based on global mean surface temperature (GST) pseudo-observations taken from the CNRM-CM6-1, r4i1p1f2 historical and SSP2-4.5 simulations. Pseudo-observations are assumed to be available over the period 1850–2030, as an illustration to help distinguish between "2023" and "current year", which in this case means 2030. These pseudo-observations are shown as orange dots. The prior on the forced warming illustrates the spread of CMIP6 models (light blue line and light blue shading stand respectively for the prior mean and 5–95% confidence range). The posterior illustrates the estimated warming range given available pseudo-observations (light green line and light green shading stand respectively for the posterior mean and 5–95% confidence range). Key variables of interest in this study involve: the estimated or projected warming in 2023, 2050 and 2100, and the forced warming to date, i.e., at the end of the pseudo-observed period–here in 2030 (respectively the dark blue, blue, orange and dark red vertical bars).

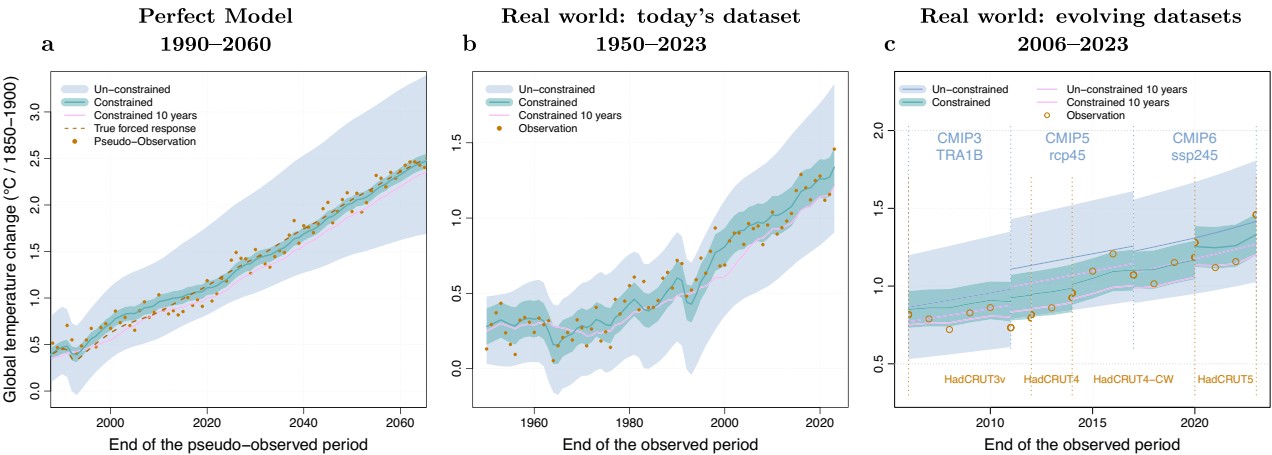

**Fig. 2 | Warming to date estimate and its changes over time.** The forced warming is estimated for the 'current' year (x-axis), via a Bayesian observational constraint, using all observations available at that time (i.e., over the period from 1850 up to the shifting date given in the x-axis), corresponding to the 1-year estimate. Calculations are done in a perfect model framework (using the CNRM-CM6-1 r4i1p1f2 historical and SSP2-4.5 simulations as pseudo-observations, panel (**a**), using real world HadCRUT5 observations as available in 2024 panel (**b**), and using datasets that were available at the date studied (including earlier versions of the HadCRUT dataset, and various CMIP generations, panel c). The observational constraint uses an ensemble of climate models as a prior ("Un-constrained", 5–95% confidence ranges, blue): the CMIP6 ensemble panels (**a**, **b**) or various CMIP ensembles panel (**c**). The result of the estimation procedure is the posterior given observations ("Constrained", best-estimate and 5–95% confidence range, green). This estimate is compared to the constrained estimate of the forced warming over the previous 10-year period ("Constrained 10-year", pink, best-estimate only). Observations or pseudo-observations are shown in orange.

over recent years[11]. If the lagged 10-year estimate is considered as an estimator of the current forced warming level, then the time-lag translates into a statistical bias, raising the question of whether this estimator is the most appropriate. The 1-year estimate avoids any time-lag, but it carries the risk of being overly influenced by the latest years observed, and thus of being too unstable. From a statistical point of view, this is a classic bias-variance compromise. Similar issues have been discussed, for example, for estimating climate normals in the context of climate change[31–33].

We show how the 1-year and 10-year warming estimates evolve as new data become available, both in climate model simulations (using a so-called perfect model approach, Fig. 2a and S1), and in observations (Fig. 2b). The uncertainty affecting these estimators can be decomposed in terms of mean and variance. We look at the variance first to assess their stability. For global mean temperature in the perfect model framework, we find a standard deviation of only 0.044 °C for the 1-year estimator (multi-model median, see "Methods" and Table S1; the smoothing applied to the forced response with our method tends to make this standard deviation relatively small). The variance of the 10-year estimator is almost the same: 0.045 °C. Regarding the bias, i.e., the average error between an estimator and the true forced warming, the 1-year estimate is almost unbiased (−0.02 °C), while the 10-year average exhibits a bias of about −0.13 °C. As a result, in terms of root-mean-square error, the 1-year estimate is much more accurate than the 10-year estimate for characterising the current forced warming of the climate system, given the strong underlying and continuing decadal warming trend. In terms of inter-annual variations, the occurrence of a particularly cold year due to internal variability typically induces a stabilisation of the 1-year estimate between year N and N + 1 – i.e., the downward revision of the estimator is offset by the average warming trend (Fig. 2b). On the other hand, the 1-year estimate has increased by 0.05 °C between 2022 and 2023, due to the very high observed GST in 2023−this increase is close to twice the current warming rate (0.25 °C/decade)[11,12,21]. Overall, the additional revision from a particularly hot / cold year is quite symmetric if the long-term trend is removed. This revision is small compared to the magnitude of the inter-annual variability that causes the revision, and it is small compared to the overall uncertainty on forced warming estimates. Even in terms of

inter-annual variations, the use of a 10-year average provides no added value since both estimators show virtually the same year-to-year variability (Fig. 2a, b).

Similar results are found on a regional scale (Figs S2, S3 and Table S2, "Methods"): the 1-year estimator exhibits a higher accuracy than the 10-year estimator. This result supports using the 1-year estimate as the most relevant estimate of the current regional warming. However, a downward revision of the estimated warming to date between years N and N + 1 occurs occasionally in this case, due to a lower signal-to-noise ratio at the regional scale.

This analysis suggests that the current warming (1-year) estimate obtained from the KCC approach is accurate (unbiased, with a modest variance) and stable. This finding is consistent with those previously reported with the Global Warming Index (GWI), e.g., "the GWI is relatively insensitive to the end date as well as short-term GMST fluctuations"[7]. It supports the use of the current year (1-year) estimate as the best indicator of the current warming level.

However, the above analysis is carried out within an idealised framework, which does not incorporate all the sources of uncertainty affecting the warming to date estimate. In the real world, observed datasets evolve as a result of progress in recovering past observations, homogenisation, and other methodological advances. For example, since 2005, 4 versions of the HadCRUT datasets[34–37] (one source of observed GST estimates) and three climate model generations[38–40] have been used (see Methods). How is the warming to date estimate affected by these changes?

Figure 2c shows a replay of the current warming estimate over the last 20 years, based on the datasets available at each point in time. This retrospective analysis shows that changes in reference datasets (observed and simulated) can induce non-negligible variations in the current warming estimate, typically from a few hundredths to almost a tenth of a degree. These sources of uncertainty can thus be larger than the variations induced by the addition of one more observed year. However, the 10-year average estimate is as sensitive to these changes as the 1-year estimate. Consequently, uncertainty related to the input datasets can be considered as mainly irreducible by the statistical method, and should be incorporated explicitly as much as possible (as done here with measurement uncertainty).

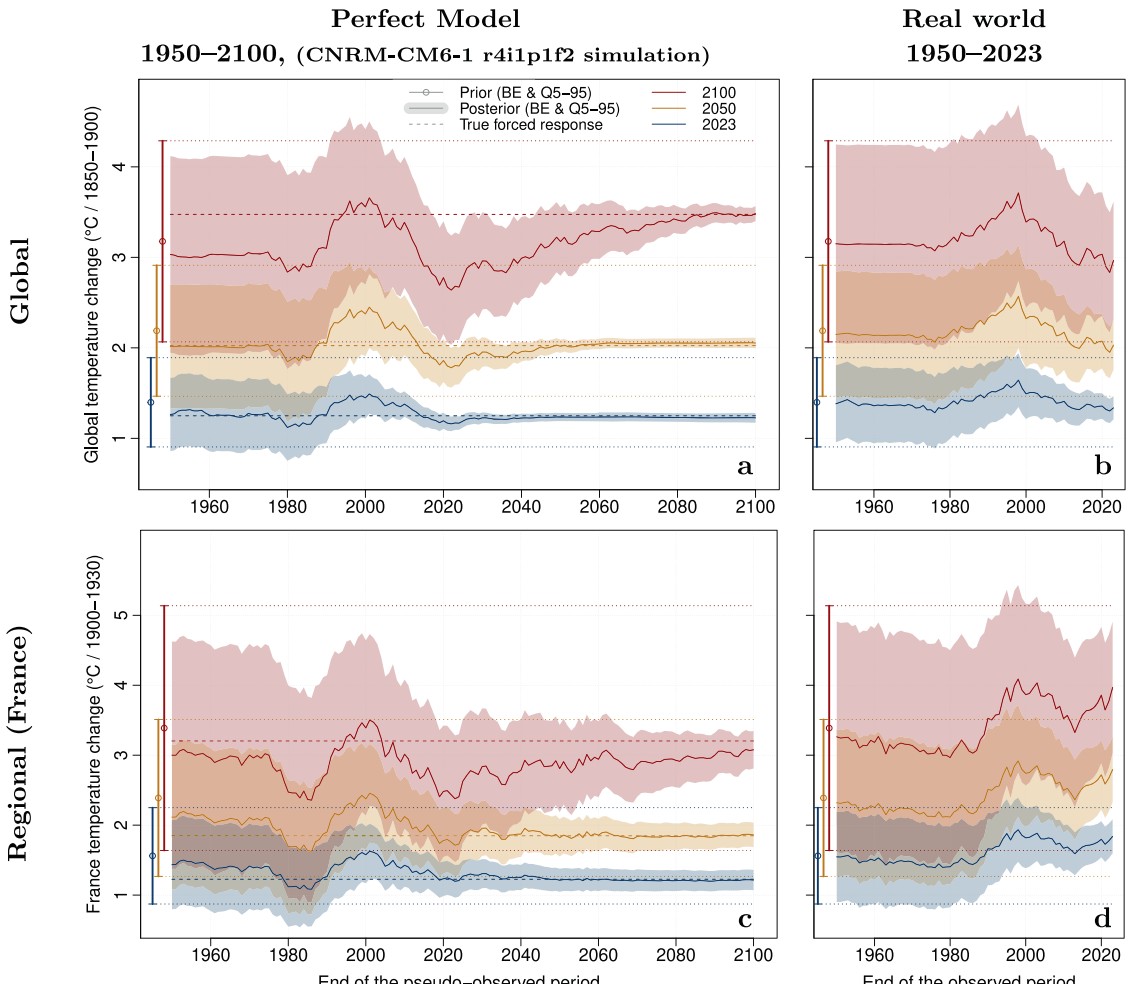

**Fig. 3 | Changes in observationally constrained projections as new observations become available.** Changes in observationally constrained projections are shown in a perfect model framework (using the CNRM-CM6-1 r4i1p1f2 historical and SSP2-4.5 simulations as pseudo-observations, (**a, c**), and in the real world (using HadCRUT5 observations, (**b, d**) All observational constraints make use of observations or pseudo-observations from 1850 up to a shifting date given in the x-axis. The analysis is replicated at the global scale (top) and over France (bottom) as an illustration of a regional scale. Observationally constrained projections are shown for 2023 (dark blue), 2050 (orange) and 2100 (dark red). All confidence ranges are 5–95%, with the best-estimate (median) shown as a solid line. Confidence ranges on the left margin of each panel correspond to the CMIP6 unconstrained confidence ranges. In the perfect model framework, horizontal dashed lines show the best estimate of the warming in 2050 and 2100 in the CNRM-CM6-1 model, as derived from the full ensemble of 10 SSP2-4.5 members from this model.

## Constrained projections

To assess the benefit of annual updates on projections, we illustrate how estimates of future forced warming and their uncertainty range evolve as new observed years become available. A first assessment is made in a perfect model framework that uses model simulations covering the entire 21st century as pseudo-observations (Fig. 3a and S1). Then, the same assessment is made using real-world observations, available up to 2023 (Fig. 3b). The two analyses are performed using an intermediate emissions scenario SSP2-4.5, and so do not cover scenario uncertainty[37]. Importantly, our focus remains on the forced response: although recent observations are used, we do not seek to deliver "initialised" projections, nor predict a specific state of internal variability.

Both analyses suggest that, in the current era, the estimation of future warming becomes continuously more accurate by incorporating new observed data. We find that adding new observed data helps to refine the projections, and that, over time, constrained ranges come closer to the 'true' response of each model—although convergence is less good for models which appear as outliers within the whole CMIP6 range (Fig S4). The constrained projected warming range for a given date continues to shrink even after that date. This simply shows that

estimation of the forced warming for a given date benefits from observations made after that date. Moreover, the added value of one individual year remains modest for projections: in Fig. 3a, in the early 21st century, the best-estimate 2100 warming typically shifts by +/−0.07 °C (mean absolute error) if one additional year is added, and the corresponding confidence interval shrinks by 1.5%. The modest revision from one year to the next suggests that the constrained estimate is relatively stable with respect to inter-annual observed variability, with no overfitting to the last observed values. Our results also show that uncertainty associated with 2050 and 2100 warming, in response to an intermediate SSP2-4.5 scenario, has already decreased significantly over the last 2 decades (Fig. 3b, d), consistent with earlier expectations[41]. In terms of the global average, there was a very gradual downward revision during the global warming slowdown period, between 1998 and 2014[10], with the 2100 SSP2-4.5 warming best-estimate shifted by −0.8 °C between these two dates, followed by a +0.2 °C revision induced by the strong El Nino episode in 2015–16, and then a further downward revision (Fig. 3b).

Carrying out the same analysis at the regional scale leads to qualitatively similar results regarding the gradual improvement in accuracy of the constrained projections (Fig. 3c, d and S5). However,

due to a lower signal-to-noise ratio, the convergence is slower, and uncertainty on the amount of forced warming at a given date (e.g., 2050) remains substantial even after that date.

These results suggest that updating climate projections on an annual basis is worthwhile in order to provide the most accurate estimate of future warming.

## Discussion

Taken as a whole, our results provide support for updating estimates of current and future (observationally constrained) warming on a yearly basis by incorporating the latest available observations. Our results also demonstrate the robustness of estimating the forced warming to date for the current year, specifically, thus avoiding any time-lag. This is particularly relevant to provide an up-to-date picture of forced global warming. These findings support the provision of regular updates of present and future warming, on both global and regional scales, consistent with other indicators (e.g., emissions, carbon cycle, radiative forcing). This study also highlights the benefits of using observational constraint methods to provide a consistent, seamless estimate of past, present and future warming. Unlike alternative proposed approaches[5], our method accounts for (CMIP) modelling uncertainty, and does not require regular updates of climate simulations in order to update this estimate, ensuring traceability. These properties make it particularly attractive.

These results also bring new challenges and new opportunities.

This study uses scenario simulations (SSP2-4.5) after the end of the historical CMIP6 simulations, i.e., after 2014. A first challenge concerns the forcings used in numerical simulations over recent years. The forcings prescribed in the scenario simulations may differ substantially from those in the real world over the same period, in particular for short-lived climate forcers and natural forcings. Two recent examples are the eruption of the Hunga-Tonga volcano[42,43], or the decline in shipping aerosol emissions[11,44,45]–noting also that anthropogenic aerosol emissions as a whole remain fairly uncertain[46]. If a major volcanic eruption occurs, this discrepancy could become large. Then, the method used in this article would have to be adapted to account for new forcing estimates, as done with simple climate models[7]. As a result, accurately estimating the human influence on climate and the current state of the system would also depend on progress in estimating these forcings.

Our study demonstrates the value of regular updates and the robustness of the results using our method within a perfect model framework based on CMIP6 models. However, our method assumes that "models are statistically indistinguishable from the truth", and might deliver biased results if all models were consistently biased. Method evaluation suggests that the presence of models with a high climate sensitivity within the CMIP6 ensemble[44] does not hamper the capacity of the method to properly estimate the forced response. Other recent studies have highlighted the difficulties current models have in realistically simulating some features of observed climate change, such as the warming pattern[47] or the trend in Earth's energy imbalance[48], with possible consequences for the use of observational constraints[49–52]. This is part of a general debate on the realism of CMIP models and the potential role of internal variability or some feedbacks in explaining some discrepancies between models and observations, a debate that has not yet been settled[53–55]. Eventually, these discrepancies could make observational constraints less effective, e.g., if the magnitude of observed GST internal variability were revised upward, or if the statistical relationship between past and future warming were weaker than current models suggest. Ultimately and qualitatively speaking, the value of annual updates and their relative stability as new observations are taken into account, as shown here, will not necessarily be challenged.

In terms of opportunities, the regular integration of observed data is similar to data assimilation. It is carried out here in a specific context

since it does not require simulations to be rerun. Critical methodological progress could be made by implementing state-of-the-art data assimilation techniques in the climate context. The updating of constrained climate projections could easily be extended to update estimates of climate sensitivity, while remaining aware of limitations related to the pattern effect uncertainty. Lastly, regular updates as described in this study could benefit projections of many other variables, including variables for which observational constraints are already effective (e.g., ocean heat content, sea level, Arctic sea ice extent[56]), and new variables, as climate change strengthens (e.g., hydroclimate, atmospheric circulation, etc).

The integration of the latest available observations in the estimation of projections, as proposed here, reflects a warming world in which observations are providing increasingly accurate information on the features of climate change. In this context, regular updates of the physical climate system response could usefully complement indicators of human influence itself (e.g., inventories of emissions and sinks of well-mixed greenhouse gases and short-lived climate forcers, as well as current national mitigation policies and nationally determined contributions, which are also assessed on an annual basis), in order to advance estimates of the expected warming over the next few years/decades or remaining carbon budgets[57]. Such updates would also be extremely helpful for authors of the forthcoming Intergovernmental Panel on Climate Change AR7 reports, by providing solid, near-real-time information aggregating multiple lines of evidence.

## Methods
### Statistical methods
**Observational constraint.** This study relies on the Kriging for Climate Change (KCC) statistical method, first introduced to constrain Global mean Surface Temperature (GST)[22], and subsequently extended to tackle regional to local temperatures[23,24]. This method seeks to estimate the forced warming, both in the past (1850 to present) and the future (present to 2100, for a given emission scenario), through an observational constraint approach that combines model data and observations. Here we provide a summary of the key elements of this method, but refer to the corresponding publications[22,24] for a comprehensive description.

The KCC method is basically a Bayesian estimation method, and as such involves the formulation of an a priori distribution of the forced response. It works in three steps, and is described here for a given emission scenario.

First, the forced response of a range of climate models is estimated for the period 1850–2100. This calculation is based on the response to natural forcings, as estimated using a simple climate model[58], as well as a smoothing procedure for the remaining human-induced forced response.

Second, the sample of the forced responses from available CMIP6 climate models is used to formulate a Gaussian prior of the real-world forced response, under the assumption that "models are statistically indistinguishable from the truth". Denoting the time-series of the real-world forced response over 1850–2100 as $\mathbf{x}$ (i.e., $\mathbf{x}$ is a vector of length 251), the prior distribution for $\mathbf{x}$ is:

$$\pi(\mathbf{x}) = N(\boldsymbol{\mu}_m, \boldsymbol{\Sigma}_m) \qquad (1)$$

where N stands for the multivariate Gaussian distribution, and $\boldsymbol{\mu}_m$ and $\boldsymbol{\Sigma}_m$ are taken as the empirical mean and covariance of the CMIP6 models' forced response. As a result, this prior includes physical information from climate models and is representative of model uncertainty ($\boldsymbol{\Sigma}_m$).

Third, observations are used to derive a posterior distribution of the past and future forced response given observations, in a Bayesian

way. We assume that

$$\mathbf{y} = \mathbf{H}\mathbf{x} + \varepsilon, \text{ with } \varepsilon \sim N(\mathbf{0}, \Sigma_y) \qquad (2)$$

where $\mathbf{y}$ denotes observations (a vector from, e.g., 1850 to 2023), $\mathbf{x}$ is still the forced response from 1850 TO 2100, $\mathbf{H}$ denotes an observational operator (not all years in $\mathbf{x}$ have a corresponding observation in $\mathbf{y}$), $\boldsymbol{\varepsilon}$ denotes random noise (a vector) corresponding to internal variability and measurement errors, and $\Sigma_y$ denotes its covariance matrix. In this way, the raw observations $\mathbf{y}$ are considered to be observations of the forced response, subject to some uncertainty (mainly arising from internal variability and measurement errors). Given these assumptions, the posterior can be derived as

$$p(\mathbf{x}|\mathbf{y}) \sim N(\boldsymbol{\mu}_p, \Sigma_p) \qquad (3)$$

with the posterior mean $\boldsymbol{\mu}_p$ and the posterior covariance matrix $\Sigma_p$ available in closed forms.

As a result, this method provides observationally constrained estimates of past, present and future forced warming (i.e., the entire vector $\mathbf{x}$), including projections.

The analyses presented here are identical to those previously published for GST[22] (except for the larger set of CMIP6 models used), and France[23]. This approach differs from the Global Warming Index (GWI)[7] in terms of both the physical models used (CMIP models here vs a 2-box Energy Balance Model), and the statistical procedure used to fit observations (Bayesian inference here vs linear regression as used in optimal fingerprinting).

It is important to notice that, despite the use of recent observations, KCC (like other observational constraint methods) does not produce "initialised" projections. It is not intended to account for or predict a specific state of internal variability, such as ENSO variability. Observations are only used to improve the estimation of the forced response.

## Method evaluation

The reliability of the KCC method has been evaluated in several ways.

First, for GST only, a perfect model framework was used to evaluate the method for future projections[22]. This evaluation suggested that the method was not overly confident, as the coverage probability (i.e., the frequency with which the estimated constrained range contains the true value of interest) was estimated to be 91% in 2100 over a sample of 66 CMIP6 simulations. This is very consistent with the expected nominal value of 90%. The evaluation also illustrated the fact that models with a particularly high (eg, hot models) or low warming were correctly predicted. This result was obtained using all CMIP6 models to derive the prior, and so suggests that the presence of hot models does not bias the outputs of our method.

Second, a similar perfect model framework was used to assess the reliability of the method for regional temperature[24]. Again, the reported coverage probabilities suggested the method was reliable and not overly confident.

Third, the specific calculations performed in the current study provide additional evidence regarding the reliability of KCC in estimating both the forced warming to date (see Figs. S1 and S3 for the global and regional scales, respectively) and the projected forced warming (see Figs. S4 and S5, again for the global and regional scales, respectively). These figures show a perfect model evaluation: one simulation (historical and SSP2-4.5) from one CMIP6 model is used as a pseudo-observation, and the KCC method is applied to it. They illustrate how our estimates change over time as new pseudo-observed years become available. The estimated range is then compared to the best possible estimate of the particular model's forced response – derived using all available members (ensemble mean), the pseudo-observations for the entire 1850–2100 period, and a smoothing

method. Consistent with previous evaluations, the results show that the KCC method is able to correctly estimate both the current and future forced responses, as the "true" value lies within the estimated confidence range in most cases. This finding applies to a broad range of CMIP6 models, including some with a relatively low or high sensitivity, and some with relatively low or high responses to the aerosol forcing. Additionally, this finding was made using all CMIP6 models in the prior, including the hottest ones.

Finally, these various lines of evidence show that KCC is a reliable method to estimate the present and future forced response, provided that key assumptions are satisfied, in particular "models are statistically indistinguishable from the truth", meaning that all models are not consistently biased.

## Bias and variance of the warming to date estimators

The statistical properties (bias, standard deviation, root mean square error) of the 1-year and 10-year estimators of the warming to date are shown in Tables S1 and S2. This simple calculation is based on the analysis of the perfect model framework and on all the model runs analysed in Fig. 2a and S1. First, we derive a best estimate of the forced response in each CMIP model. This is done by averaging over all available members (the ensemble size varies across models, but all of the models used have 7 members or more of historical simulations and SSP2-4.5 scenarios), and then applying a filtering procedure[22] in which the anthropogenic response is smoothed over time to eliminate most of the remaining internal variability. This best-estimate is then considered as the *true* forced response, and is compared to the estimated warming to date using the pseudo-observations from one specific member (global scale, as illustrated in Fig. 2a and S1) or 4 members (regional scale, as illustrated in Fig S3 with one single member; the number of members used to calculate the bias and variance is increased in order to ensure the robustness of the results, despite the lower signal-to-noise ratio at the regional scale). The difference between these two quantities is considered as the error in the warming to date estimator. From the sample of errors over the period 2015–2100, we can easily derive indicators such as bias, standard deviation, and RMSE.

## Observed data

The observed global temperature data are taken from the infilled version of the HadCRUT5[35] dataset; only the annual and global means are used. This dataset covers the period 1850-2023. HadCRUT5 measurement errors are estimated using the set of 200 members provided by this dataset, assuming Gaussian error. Following recent studies[11,12], we assume that the Global Surface Temperature (GST, mixing surface atmospheric temperature over land and sea surface temperature over oceans) warming assessed in this dataset is representative of GSAT (surface atmosphere temperature everywhere) changes, although this is still a matter of debate.

In Fig. 2c shows the results of a retrospective calculation based on the data available for each year. We use various earlier versions of this dataset:

- From 2006 to 2012, we used the HadCRUT3v dataset, which is a statistically adjusted version of HadCRUT3[36]. This dataset combines land (CRUTEM3) and ocean (HadSST2) data at a 5 × 5° resolution since 1850.
- From 2012 to 2014, the HadCRUT4 dataset[34]. HadCRUT4 combines land surface temperature data from CRUTEM4 and sea surface temperature data from HadSST3. HadCRUT4 measurement errors are estimated from a set of 100 equiprobable realisations.
- From 2014 to 2020, a new version of HadCRUT4 was used, based on a better reconstruction and fusion of land and ocean surface temperatures. This version is referred to as HadCRUT4-CW[37]. Similar to HadCRUT4, this dataset provides an ensemble of 100 equiprobable members to assess measurement uncertainty.

– From 2020 onwards, the HadCRUT5 dataset, i.e., the current version, was used. As this version was published in 2021, it is used here from the year 2020 onwards. This is based on the assumption that the analysis is carried out in the year following the last year of observations used, similar to the approach of Forster et al. (2023).

The results shown in Fig. 2c suggest that revisions of these observed datasets all resulted in small breaks in the estimated warming. Over the last 20 years, all revisions have been upwards.

The observed temperature data for France comes from Météo-France[23]. These are monthly temperatures over mainland France since 1899, derived as the average of observations from 30 homogenised stations evenly distributed across the country.

## Model data

All analyses but Fig. 2c are based on a sample of 45 CMIP6 models[40]—see the detailed list below. To improve the estimation of the forced response, we use historical and SSP2-4.5 scenario simulations and take the average over all available members. For Fig. 2c, we additionally use the previous CMIP3[38] and CMIP5[39] generations. In each case, we use intermediate emission scenarios to extend historical simulations (consistent with the use of SSP2-4.5 in CMIP6): A1B for CMIP3, and RCP4.5 for CMIP5. Results in Fig. 2c suggest that the transitions from CMIP5 to CMIP6 was very smooth in terms of estimating the current warming, unlike the previous transition (from CMIP3 to CMIP5).

The processing of model data involves the following steps. The mean temperature of each model is calculated by considering the annual mean surface air temperature (SAT). For global analyses, we compute the Global mean SAT (GSAT) and assume that it is equal to GST. To illustrate regional analyses, we use temperature data over France, which are interpolated onto a 0.25° x 0.25° grid covering mainland France, and then averaged over continental grid-points (consistent with Ribes et al., 2022). At both the global and regional scales, the subsequent processing of the model data is consistent with that described in the reference papers describing the statistical method[22,23], which involves in particular a smoothing procedure to reduce internal variability.

The following CMIP6 models (SSP2-4.5 scenario) were used for GST analyses (Figs. 1, 2, 3a,b, S1 and S4; 45 models): ACCESS-CM2, ACCESS-ESM1-5, AWI-CM-1-1-MR, BCC-CSM2-MR, CAMS-CSM1-0, CAS-ESM2-0, CESM2, CESM2-WACCM, CIESM, CMCC-CM2-SR5, CMCC-ESM2, CNRM-CM6-1, CNRM-CM6-1-HR, CNRM-ESM2-1, CanESM5, CanESM5-CanOE, EC-Earth3, EC-Earth3-CC, EC-Earth3-Veg, EC-Earth3-Veg-LR, FGOALS-f3-L, FGOALS-g3, FIO-ESM-2-0, GFDL-CM4, GFDL-ESM4, GISS-E2-1-G, GISS-E2-1-H, HadGEM3-GC31-LL, IITM-ESM, INM-CM4-8, INM-CM5-0, IPSL-CM6A-LR, KACE-1-0-G, KIOST-ESM, MCM-UA-1-0, MIROC-ES2L, MIROC6, MPI-ESM1-2-HR, MPI-ESM1-2-LR, MRI-ESM2-0, NESM3, NorESM2-LM, NorESM2-MM, TaiESM1, UKESM1-0-LL.

The following CMIP6 models (SSP2-4.5 scenario) were used for regional analyses (Fig. 3c, d, S2, S3 and S5; 27 models; same models as in Ribes et al[23].): ACCESS-CM2, ACCESS-ESM1-5, AWI-CM-1-1-MR, CAMS-CSM1-0, CanESM5-CanOE, CanESM5, CESM2, CESM2-WACCM, CMCC-CM2-SR5, CNRM-CM6-1-HR, CNRM-CM6-1, CNRM-ESM2-1, EC-Earth3-Veg, FGOALS-f3-L, FGOALS-g3, GISS-E2-1-G, INM-CM4-8, IPSL-CM6A-LR, MIROC6, MIROC-ES2L, MPI-ESM1-2-HR, MPI-ESM1-2-LR, MRI-ESM2-0, NorESM2-LM, NorESM2-MM, TaiESM1, UKESM1-0-LL.

The following CMIP5 models were used (Fig. 2c; RCP4.5 scenario; 31 models): BNU-ESM, CESM1-CAM5, FGOALS-s2, MRI-CGCM3, MIROC-ESM, MIROC5, MIROC-ESM-CHEM, GISS-E2-R, GISS-E2-H, MPI-ESM-MR, MPI-ESM-LR, CanESM2, CNRM-CM5, EC-EARTH, FIO-ESM, bcc-csm1-1-m, bcc-csm1-1, NorESM1-ME, NorESM1-M, CSIRO-Mk3-6-0, CCSM4, IPSL-CM5A-LR, IPSL-CM5A-MR, ACCESS1-3, ACCESS1-0, CESM1-BGC, GISS-E2-R-CC, GISS-E2-H-CC, inmcm4, CMCC-CMS, IPSL-CM5B-LR.

The following CMIP3 models were used (Fig. 2c; AIB scenario; 23 models): BCCR, CCCMA0, CCCMA, CNRM, CSIRO0, CSIRO, GISSA, HCM, INM, IPSL, MRI, CCSM, ECHO, FGOALS, GFDL0, GFDL, GISSH, GISSR, HGEM, INGV, MIROC0, MIROC, MPI, PCM.

## Data availability

The pre-processed CMIP6 data used to perform this study are available under the following https://doi.org/10.5281/zenodo.14859923. Key results generated by our observational constraint calculation are also available in the same database. The full description of observed and model raw data used in this study is provided above (Method section).

## Code availability

The kriging for climate change R package used to produce this analysis is available at https://gitlab.com/saidqasmi/KCC; https://zenodo.org/records/5233947

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

## Acknowledgements

We acknowledge the World Climate Research Programme, which, through its Working Group on Coupled Modelling, coordinated and promoted CMIP6. We thank the climate modelling groups for producing and making available their model output, the Earth System Grid Federation (ESGF) for archiving the data and providing access, and the multiple funding agencies who support CMIP6 and ESGF. We acknowledge support by Météo-France and CNRS. We acknowledge Christophe Cassou for having prompted our interest in the issue of estimating global warming to date. This study has received funding from Agence Nationale de la Recherche–France 2030 as part of the PEPR TRACCS programme under Grant ANR-22-EXTR-0005.

## Author contributions

A.R. designed the study and wrote the paper. O.T. processed and analysed the data and produced the figures. A.R., O.T., P.F., N.G., V.M.D., J.R., R.V., and T.W. participated in the critical analysis of the results, which resulted in carrying out new analyses, and contributed to the writing of the final version of the manuscript.

## Competing interests

The authors declare no competing interests.
