## [Transparent Peer Review file · Nature Communications]

Towards annual updating of forced warming to date and constrained climate projections

Corresponding Author: Dr Aurélien Ribes

Version 0:

Reviewer comments:

Reviewer #1

(Remarks to the Author)

The paper is well-written and addresses a question that is of broad interest that seems to fit Nature Communications. The performance of an established Kriging for Climate Change (KCC) method is determined for the concept of using annually updated temperature data to estimate (with uncertainty) the forced warming through that year, and to constrain future projections. There are other published approaches that have been evaluated, including statistical fits (e.g. Clarke et al. 2022, doi: 10.1029/2020EA001082) or the “global warming index” (GWI, Hausteine et al, 2017, doi: 10.1038/s41598-017-14828-5). The submission's method is different and complements those approaches.

I particularly liked that the consequences of changes in CMIP generations or datasets were evaluated in a “retrospective” analysis, which showed that the authors considered major practical issues for implementing their method operationally.

I could support publication subject to some changes. I ask the authors to (i) clarify some Methods details and (ii) explicitly state how to interpret some results in light of the assumptions, and based on my interpretation of the assumptions some further sensitivity tests should be done to wrap up the paper nicely.

METHODS ISSUES

I had to keep Ribes et al. (2021) open side-by-side to interpret things, but a “communications” journal paper should be self-contained with regards to the major conclusions. Some details and relevant consequences should be made clearer, although you don't need to fully re-describe everything.

L305 onwards would flow better if you said something like: “First, the forced response of each model is estimated over 1850—2100 as described in the sections below.” So I would know that the “forced” calculation description will arrive.

Then on your “Second” and “Third” points in the same paragraphs, they are vague. My suggestion would be to write out Bayes' theorem and then assign each term. In my interpretation, the derivation of the prior, which seems to be assumed Gaussian with mean and covariance derived from the CMIP6 models, is very important for understanding and interpreting the paper.

ASSUMPTIONS, PRIORS, TRADEOFFS

The authors will have thought about a lot of the big questions and will understand the nuances and relevant consequences of their method. I believe there are places to share these to help readers, and perhaps add a handful of extra sensitivity tests to better understand the consequences of the assumptions which rely very heavily on the CMIP6 models.

Examples:

While you state you use the analysis version from Ribes et al. (2021), it still appears to make the assumption that from Ribes et al. (2017, in Clim Dyn) “models are statistically indistinguishable from the truth”. I found this to be useful early framing, and something similar would be helpful here. For example, as context for L86—88 where you ask if the latest observations are

“consistent”.

I also think that you should explicitly state that scenario uncertainty is ignored in your main calculations, with reference to Hawkins & Sutton (2009, doi: 10.1175/2009BAMS2607.1). It's implied in L213—216 and L251—253, but readers might miss the point for the 2100 results.

L286—296 paragraph. Elsewhere you compare with the global warming index, which uses an energy balance model while you rely on CMIP models. Doesn't that mean that your method is less able to update in response to things like “indicators of human influence (e.g., inventories of emissions and sinks...)” since you would need to wait for new CMIP runs? Or perhaps emulators (or constraints if your prior includes the histGHG or similar runs...?) could be used in place. It's not clear to me how the L286—296 paragraph claims necessarily align with the method as I understand it.

L253—255 and L262—266 lead into a “general debate on the realism of CMIP models”. Isn't a lot of this just rephrasing that there is some additional uncertainty related to the fact your prior is based on CMIP, you assume it is a reasonably random sampling of realistic models, and that “models are statistically indistinguishable from the truth”? Figure 2c probably captures some of the sensitivity to this issue (albeit also with forcing changes). The most important question for me is: what would your results do if the CMIP models were wrong in a meaningful way? Is there any way we could realise your results are wrong by some year if the models are notably wrong? In my judgment, there should be some kind of sensitivity test or analysis to touch on this point. Fully quantifying it is probably out of scope, but some tests could help indicate the level of confidence or concern we should have.

Example test: what happens if you re-derive your priors excluding either the set of highest or lowest sensitivity (TCR/ECS) models, and then try to fit with pseudo-observations derived from models that were excluded?

Alternative test: you could modify model pseudo-observations to represent a case where we are wrong about something. E.g. add large PDO swings (20+-year sine waves or pseudo-random noise with large memory to have similar behaviour?), or scaling components, e.g. to estimate a case with larger aerosol adjustments and therefore stronger temperature changes to aerosol.

Doing tests like the above to evaluate the likely effect on your annual estimates should be relatively little work and would address points that you yourselves raised as particularly important in the discussion.

Specific minor comments

L322: “have more than 7 members” – I assume this means ≥ 8 ? Very minor point but it would avoid questions if you said “8 or more” in that case.

L327—329: “or 4 members (regional scale)” – could you specify what this part means? My interpretation is that you averaged regional time series from four models to get your y pseudo-observations. Which seems a bit strange

Reviewer #2

(Remarks to the Author)

My synopsis: A key element of this approach is that the individual observations for a given year will most likely have the same characteristics to provide a new updated estimate of the next year's estimate. This is not very different from the data assimilation products. If it works well, we would expect that the annual update would account for internal variability. The sophistication of data assimilation tools should provide outliers that could be correct but would have small impact on the expected state of the climate. A good test would be to identify the large SST anomalies such as ENSO events or large anomalies in the high latitudes (both land and ocean).

line 47: goes -> changes

line 53: The forced response needs to be assessed for its own internal variability. Additionally, the scaling of the strong local responses cannot be averaged out when providing such forecasts. The regional climate depends on them.

line 102-103: I would put “i.e., the amount of forced warming estimated for the current year” in parentheses and put “a robust indicator” before it.

Line 108: KCC = Kriging for Climate Change

I agree that the past, present, and future warming can use the KCC filter seamlessly, but the characteristics of the filters depend (strongly) on the composition of the different noise properties. This needs to be highlighted here.

Line 134: How do the resulting uncertainties depend on the 10y and 1y cases? Would regional cases be more informative for the relevant regional areas? (both ocean and land).

line 137: Wouldn't 1-yr estimates still be smoothed out given the annual approach to the constraints?

line 143: Need to specify the noise characteristics for the estimates used to provide the warming estimates based on the observing systems.

line 148: Given the 4-7yr ENSO characteristic timescales, using the older data would be essentially generating two 5-year periods for the smoothing properties. If the ENSO events are really getting longer gaps, how would this influence the future events?

line 151: see the last comment.

line 159-161: . . . and yet, the exceptions are what confound the canonical cases.

line 201-202: This seems to state the status quo is basically running the show.

line 222 paragraph: The differences in the regional aspects would be connected to the heat content as well. It might be interesting to consider east coasts and west coasts of continents to quantify if any significant changes emerge.

line 228-229: How does an estimate for a new ENSO influence the next year's projection? Are there biases in the CMIP6 that limit the variances? I.e., are the models not showing enough variance for an upcoming ENSO?

line 236-237: Does this mean that only the current year is used, not the next year?

line 262-266: What would happen if ECMWF did the same 1-year projection? Does the data assimilation from ECMWF provide similar estimates?

line 301: I see that there could be more information in the early part of the paper to clearly state it is using the available data to estimate the "climate" year from the assimilated data.

line 375-376: I would be more interested in the OHC data for the upper 200m of the ocean. For the surface land temperature, I'm not sure if it would benefit from having a heat content approach. The memory in the heat content would be important, for both land and ocean..

Figure 1: "constraint" in the figure is misspelled for the pseudo-observation. Label for "Current year climate estimation" is not really climate, rather it is the annual mean. I think it is better to separate "climate" as a multi-year average as compared to the "annual mean". This doesn't impact the message from my perspective.

Line 460-462: I don't understand why the 10-yr constrained data are biased low. Shouldn't the 10y mean be centered rather than shown at the end of the decadal period?

Figure 3 caption: In Fig 3, I'm struggling to understand why the global and regional figures show the oscillation starting in the mid-70s. This seems to show a lag response that starts ringing with data constraints. I'm puzzled by the jump in France at 1995 and same with the Global at 1975. What is setting these shifts?

line 475: perfect is misspelled.

Reviewer #3

(Remarks to the Author)

Review of Ribes et al., Towards annual updating of forced warming to date and constrained climate projections, manuscript number NCOMMS-24-32900.

The submitted paper uses a "perfect model" to argue for the value of providing regular updates to projections of the rise in Global mean Surface Temperature (GST) that are a highlight of Physical Science Basis components of IPCC reports. They make use of output from 8 climate models in the analysis, and they also assess the multi-model mean of 1-yr and 10-yr estimator of the rise in GST.

Simply, I fail to see the value of the content of this manuscript, especially for a high-profile journal such as Nature Communications. Critical issues with climate models such as the so-called "hot model bias" (e.g., Hausfather, Z., Marvel, K., Schmidt, G. A., Nielsen-Gammon, J. W. and Zelinka, M. (2022) 'Climate 1022 simulations: recognize the 'hot model' problem', Nature, 605(7908), pp. 26-29 and references therein) are not addressed. I find the text on lines 143 to 161 to be confusing at best, disingenuous at worse. It is a long-standing problem in the community that the CMIP6 models tend to "run hot", most likely due to the tendency of the real-world to be in a more La Nina like state over time than is simulated by these models (e.g., Weaver, Clark, et al. "Comparison of proxy-Shortwave Cloud albedo from SBUV observations with CMIP6 models." Journal of Climate 37.11 (2024): 3093-3116 and references therein). There is concerted effort in the climate community to understand the anomalously high GST in 2023 in papers such as:

Esper, J., Torbenson, M. and Büntgen, U. (2024) '2023 summer warmth unparalleled over the past 2,000 969 years', Nature,

Schmidt, G. 2024. Climate models can't explain 2023's huge heat anomaly — we could be in uncharted 1199 territory. Nature.

Yuan, T., Song, H., Oreopoulos, L., Wood, R., Bian, H., Breen, K., Chin, M., Yu, H., Barahona, D., Meyer, K. 1276 and Platnick, S. (2024) 'Abrupt reduction in shipping emission as an inadvertent geoengineering 1277 termination shock produces substantial radiative warming', Communications Earth & Environment, 5(1), 1278 pp. 281.

none of which are cited. Simply, this paper falls very short of the standard I envision for Nature Communications, and I honestly feel the manuscript will be hard-pressed to review well in a lower profile journal.

Version 1:

Reviewer comments:

Reviewer #1

(Remarks to the Author)

The authors have adequately responded to my comments and I support publication. The paper is now self-contained and I feel like I have a sufficiently complete understanding of what was done to interpret the results.

I do not believe I need to see this again, although I have a couple of suggestions to refine the methods.

For the future it would be interesting to see if things change with more sophisticated accounting for structural dataset errors, the effect of structural biases in model sampling, or scenario error.

MINOR METHODS COMMENTS

- L351: Methods equation. You use "y" for the observation (forced+ICV) but then " Σy " to refer to the covariance *only in the ICV part*. Suggest changing to something like " Σy_{ICV} " for clarity.

- L360: "therefore including projections – the entire vector x". If I'm reading correctly I think you mean "of the entire vector x". Please check.

- L335-338 is an example of some of the clunkier writing in the new Methods text, with three versions of the word "estimate" in it. I think readers would appreciate it if the authors skimmed once again through the Methods to bring the language and its flow up to the standards of the rest of the paper.

Reviewer #2

(Remarks to the Author)

After reading the R2 version of the manuscript, I am satisfied with the revision. This paper has been revised is very I have also read through the two additional reviews. From my perspective, I am ready to accept the current version of the manuscript for publication. If needed, I would be available to review a third version of the manuscript.

REVIEWER COMMENTS

Reviewer #1 (Remarks to the Author):

The paper is well-written and addresses a question that is of broad interest that seems to fit Nature Communications. The performance of an established Kriging for Climate Change (KCC) method is determined for the concept of using annually updated temperature data to estimate (with uncertainty) the forced warming through that year, and to constrain future projections. There are other published approaches that have been evaluated, including statistical fits (e.g. Clarke et al. 2022, doi: 10.1029/2020EA001082) or the “global warming index” (GWI, Haustein et al, 2017, doi: 10.1038/s41598-017-14828-5). The submission’s method is different and complements those approaches.

I particularly liked that the consequences of changes in CMIP generations or datasets were evaluated in a “retrospective” analysis, which showed that the authors considered major practical issues for implementing their method operationally.

We thank the reviewer for these positive comments.

I could support publication subject to some changes. I ask the authors to (i) clarify some Methods details and (ii) explicitly state how to interpret some results in light of the assumptions, and based on my interpretation of the assumptions some further sensitivity tests should be done to wrap up the paper nicely.

As mentioned above (and detailed below), we have implemented all the requested revisions, including providing details about the method and deepening the discussion on interpretation and assumptions.

METHODS ISSUES

I had to keep Ribes et al. (2021) open side-by-side to interpret things, but a “communications” journal paper should be self-contained with regards to the major conclusions. Some details and relevant consequences should be made clearer, although you don’t need to fully re-describe everything.

We have deeply revised the method description, which was indeed not enough detailed in the previous version of the manuscript.

L305 onwards would flow better if you said something like: “First, the forced response of each model is estimated over 1850—2100 as described in the sections below.” So I would know that the “forced” calculation description will arrive.

While rewriting this whole paragraph, we have added information on how the forced response is derived. The paragraph is much more self-contained now, and provides clarity.

Then on your “Second” and “Third” points in the same paragraphs, they are vague. My suggestion would be to write out Bayes’ theorem and then assign each term. In my interpretation, the derivation of the prior, which seems to be assumed Gaussian with mean and covariance derived from the CMIP6 models, is very important for understanding and interpreting the paper.

We have followed this suggestion and introduced an explicit mathematical notation, enabling us to write out how Bayes' theorem is used in this method. Thank you for this suggestion to help readers understand our method.

ASSUMPTIONS, PRIORS, TRADEOFFS

The authors will have thought about a lot of the big questions and will understand the nuances and relevant consequences of their method. I believe there are places to share these to help readers, and perhaps add a handful of extra sensitivity tests to better understand the consequences of the assumptions which rely very heavily on the CMIP6 models.

We agree with this general comment about previous use / thinking about the KCC method.

Examples:

While you state you use the analysis version from Ribes et al. (2021), it still appears to make the assumption that from Ribes et al. (2017, in *Clim Dyn*) “models are statistically indistinguishable from the truth”. I found this to be useful early framing, and something similar would be helpful here. For example, as context for L86—88 where you ask if the latest observations are “consistent”.

This is correct: our method does rely on the “models are statistically indistinguishable from the truth” paradigm. We have added it explicitly in the presentation of the method, and more specifically in the description of the Bayesian prior (i.e., where this assumption comes into play).

I also think that you should explicitly state that scenario uncertainty is ignored in your main calculations, with reference to Hawkins & Sutton (2009, doi: 10.1175/2009BAMS2607.1). It's implied in L213—216 and L251—253, but readers might miss the point for the 2100 results. We now explicitly state that our projections do not consider scenario uncertainty, at the end of the first paragraph of the *Constrained projections* section. The reference to Hawkins & Sutton was also added.

L286—296 paragraph. Elsewhere you compare with the global warming index, which uses an energy balance model while you rely on CMIP models. Doesn't that mean that your method is less able to update in response to things like “indicators of human influence (e.g., inventories of emissions and sinks...)” since you would need to wait for new CMIP runs? Or perhaps emulators (or constraints if your prior includes the histGHG or similar runs...?) could be used in place. It's not clear to me how the L286—296 paragraph claims necessarily align with the method as I understand it.

Our point here was to stress that annual updates of the human influence on climate are already made for some variables / indicators, as “inventories of emissions and sinks”. This study is proposing an extension.

The point about our CMIP-based method being less able than others, particularly EBMs, to account for revised forcing, is indeed relevant. This limitation is discussed in the third paragraph of the method (“*This study uses...*”), and we slightly revised this paragraph to make the comparison with EBMs clearer ; in particular the fact that EBMs can more easily account for revised forcing estimates than our methodology.

L253—255 and L262—266 lead into a “general debate on the realism of CMIP models”. Isn’t a lot of this just rephrasing that there is some additional uncertainty related to the fact your prior is based on CMIP, you assume it is a reasonably random sampling of realistic models, and that “models are statistically indistinguishable from the truth”? Figure 2c probably captures some of the sensitivity to this issue (albeit also with forcing changes). The general point about “realism of CMIP” is correct. We are indeed discussing this possible lack of realism, in particular due to forcing issues, or a wrong forced warming pattern. The wording of the paragraphs pointed out has been slightly revised to make this point clearer.

The most important question for me is: what would your results do if the CMIP models were wrong in a meaningful way? Is there any way we could realise your results are wrong by some year if the models are notably wrong? In my judgment, there should be some kind of sensitivity test or analysis to touch on this point. Fully quantifying it is probably out of scope, but some tests could help indicate the level of confidence or concern we should have.

Example test: what happens if you re-derive your priors excluding either the set of highest or lowest sensitivity (TCR/ECS) models, and then try to fit with pseudo-observations derived from models that were excluded?

These two comments / suggestions are about method evaluation, which was only superficially discussed in the original manuscript. In order to better describe and discuss the reliability of the KCC method, we have added a full paragraph about “method evaluation” in the revised manuscript. This paragraph provides a summary of all analyses made in previous publications and in the current manuscript, and addresses the reviewer’s point.

The reviewer appropriately suggests testing the ability of KCC to estimate the response of one particular model, potentially a high-ECS one, while using other models to build the prior. Such tests were implemented in two previous studies, and some are also implemented in the current one, but were not sufficiently discussed in the previous version of this manuscript. In the original KCC paper (Ribes et al., 2021), KCC was tested in a perfect model framework. We took simulations in turn from each model as pseudo-observations, applied the method to all the other models except that model, and then evaluated how well the method predicted future warming in the withheld model. The set of models used as pseudo-observations included models with high ECS and models with low ECS. As a result, the method was reported to have a correct coverage probability (the estimated 90% confidence range contains the true value 90% of times). KCC was also reported to correctly estimate the forced response for models with high or low ECS. Similar findings were obtained at the regional scale (Qasmi and Ribes, 2022). Lastly, our figures S1, S3–S5 also provide results of such analyses, with a focus on the estimated warming to date (S1 and S3), and the projected forced warming (S4 and S5). All these tests and results about the method evaluation are now discussed in the manuscript (Method evaluation sub-section).

That being said, we did not attempt to use KCC in a context where the considered prior is strongly biased with respect to the “truth”, such as removing all hot models and trying to predict the response of one of them, as the reviewer suggests. What we did was limited to excluding one single model, but because there are several high-ECS models in CMIP6, this does not remove all high-ECS models, and so is a weaker test. If such a test was implemented, we do not expect KCC to be able to estimate the correct forced response. Stated differently, our method indeed assumes that “models are statistically indistinguishable from the truth”, and if that assumption really fails (meaning all models are substantially

biased in a consistent way), KCC could provide biased estimates too. This limitation is now clearly mentioned in the method description (end of the new model evaluation paragraph), and in the main text (fourth paragraph of the discussion section).

Alternative test: you could modify model pseudo-observations to represent a case where we are wrong about something. E.g. add large PDO swings (20+-year sine waves or pseudo-random noise with large memory to have similar behaviour?), or scaling components, e.g. to estimate a case with larger aerosol adjustments and therefore stronger temperature changes to aerosol.

Regarding this alternative test: we feel adding a really wrong/missed signal might not be that useful to evaluate the method, for two reasons.

First, if the method is applied to pseudo-observations containing a really wrong signal, then the method assumption that models are “statistically indistinguishable from the truth” would be violated, and so the method would be expected to provide inaccurate estimates in such a case. To stress this, the revised manuscript includes an explicit mention of the method limitation (as discussed above).

Second, the available set of climate models already provides some evaluation of the reviewer's suggestion – physically plausible ones, as it is provided by state-of-the-art climate models. Regarding low-frequency variability : IPSL is known to exhibit a very large, low-frequency AMOC variability, affecting Europe's climate (among other regions; see eg Jiang et al., 2021; Parsons et al., 2020). The incorrect prediction of the current forced warming over France in Figure S3 (IPSL panel) might be due to this variability. Regarding a strong aerosol cooling, UKESM is known to be an example. Our evaluation suggests this only marginally affects the current forced warming estimate (see Fig S1 and S3: UKESM is correctly monitored), but substantially affects the projections (see Fig S4 and S5: before 2030 approximately, the forced warming of UKESM is underestimated, presumably because the large cooling effect of aerosols is not satisfactorily accounted for).

Jiang, W., Gastineau, G., & Codron, F. (2021). Multicentennial variability driven by salinity exchanges between the Atlantic and the Arctic Ocean in a coupled climate model. *Journal of Advances in Modeling Earth Systems*, 13(3), e2020MS002366.

Doing tests like the above to evaluate the likely effect on your annual estimates should be relatively little work and would address points that you yourselves raised as particularly important in the discussion.

We have reviewed a broad range of tests of this method and their results, which demonstrate the relevance of the KCC method for the purpose of this study. We have also more clearly mentioned the assumptions and limitations of this method in the revised manuscript, as suggested by the reviewer.

Specific minor comments

L322: “have more than 7 members” – I assume this means ≥ 8 ? Very minor point but it would avoid questions if you said “8 or more” in that case.

Corrected to “7 or more” (not “8 or or more”)

L327—329: “or 4 members (regional scale)” – could you specify what this part means? My interpretation is that you averaged regional time series from four models to get your pseudo-observations. Which seems a bit strange

The wording here was misleading, and has been revised. The Fig S3 is based on the analysis of one single member, for each climate model. But the calculation of bias and variance is based on 4 different members, for each model, in order to ensure the robustness of the results.

Reviewer #2 (Remarks to the Author):

My synopsis: A key element of this approach is that the individual observations for a given year will most likely have the same characteristics to provide a new updated estimate of the next year's estimate. This is not very different from the data assimilation products. If it works well, we would expect that the annual update would account for internal variability. The sophistication of data assimilation tools should provide outliers that could be correct but would have small impact on the expected state of the climate. A good test would be to identify the large SST anomalies such as ENSO events or large anomalies in the high latitudes (both land and ocean).

The reviewer is correct in making a parallel between our method and data assimilation. The Kriging for Climate Change method is indeed very similar to data assimilation. It includes a Kalman Filtering step, as for data assimilation. The topic of providing regular updates is also very consistent with operational data assimilation.

However, we would like to stress that the proposed method and analysis focuses on the forced response only, with no information provided about internal variability (IV). For instance, the method does not account for a particular ENSO state. The method uses an estimate of how much internal variability affects observations, but purely from their statistical analysis (ie how much variance). There is no additional information on whether or not this IV has, eg, offset or strengthened global warming over the current year.

We feel this is an important point as several comments below are related to this point.

Therefore we have added specific sentences, in the main text (1st paragraph of the Constrained projections section), and in the Method section, at the end of the “Observational constraint” paragraph.

line 47: goes -> changes

Corrected

line 53: The forced response needs to be assessed for its own internal variability.

We were a bit confused by the first sentence: “internal variability of the forced response”. We understand this as the uncertainty affecting the estimate of the forced response. If so, we believe that this uncertainty is deeply assessed in the manuscript, as all estimates are provided with uncertainty ranges, the KCC method is designed to take into account uncertainties affecting model projections and observations, and our uncertainty calculations are double checked in the new method evaluation section.

Additionally, the scaling of the strong local responses cannot be averaged out when providing such forecasts. The regional climate depends on them.

This is correct. Our study provides one single example of regional analysis (Fig 3c,d, Fig S3, S5). In this example, the local response is not averaged out with responses from other regions (what the reviewer fears, according to our understanding). The calculation for the regional forced response accounts for global and regional observations. As the regional observations are specific to each region, the “scaling of the strong responses” is also specific to each region.

line 102-103: I would put "i.e., the amount of forced warming estimated for the current year" in parentheses and put "a robust indicator" before it.

Corrected. We removed the explanation of “warming to date” , since a more accurate explanation is (and was already) given just a little below (Result section).

Line 108: KCC = Kriging for Climate Change

I agree that the past, present, and future warming can use the KCC filter seamlessly, but the characteristics of the filters depend (strongly) on the composition of the different noise properties. This needs to be highlighted here.

A sentence has been added at this place:

“This method involves mainly a Kalman filter or Kriging, and its implementation requires a careful estimation of the uncertainties affecting climate models and observations.”

Line 134: How do the resulting uncertainties depend on the 10y and 1y cases? Would regional cases be more informative for the relevant regional areas? (both ocean and land). We have revised this sentence to make the point clearer, and highlight the fact that the 10-yr estimator is being discussed here (the mention of “time-lag” in the previous version was potentially unclear). The “relative uncertainties” of the two estimators is dealt with in the subsequent sentences. Overall, the following 2 paragraphs are meant to quantify / compare the uncertainties of these 2 estimates, and draw conclusions (1-yr estimate is better than 10-yr).

Another sentence was added to address the “regional cases” information – just to make it clear that our regional results suggest using a regional 1-yr estimate.

line 137: Wouldn't 1-yr estimates still be smoothed out given the annual approach to the constraints?

The 1-yr estimate is derived from the same KCC procedures, which is now explained in the method section. This procedure involves smoothing. However, even a smoothed estimate can be influenced by particular values, and part of our manuscript focuses on quantifying this influence – and showing this remains limited. The sentence has been rephrased, in order to describe how the estimator is “influenced by” specific values.

As a complement, and in response to several comments from Reviewer 2 about this paragraph, we have added the following sentence:

Importantly, these estimators are both derived from an attribution method, aimed at separating forced response and internal variability - they are not just the observed warming over 1-yr or 10-yr.

line 143: Need to specify the noise characteristics for the estimates used to provide the warming estimates based on the observing systems.

This sentence has been revised to make it clearer. Here, we are looking at how the estimates evolve over time, not just the effect of internal variability. The following sentence was also revised, to make it clearer that this entire paragraph does investigate the “noise characteristics” (ie, the uncertainty) of these estimates.

line 148: Given the 4-7yr ENSO characteristic timescales, using the older data would be essentially generating two 5-year periods for the smoothing properties. If the ENSO events are really getting longer gaps, how would this influence the future events?

As this study focuses on estimating the forced response, without explicitly modeling the ENSO variability, a change in the ENSO spectrum would not be highly problematic. Using a smoothing technique, as proposed here, simply requires that the (internal) variability be “centered” on the (slowly moving) mean forced response, and this would still be the case, even if the ENSO variability were to change.

By the way, pieces from recent literature suggest that there is remaining uncertainty on how the ENSO is responding / will respond to climate change, see e.g.,

Maher, N., Wills, R. C. J., DiNezio, P., Klavans, J., Milinski, S., Sanchez, S. C., Stevenson, S., Stuecker, M. F., and Wu, X.: The future of the El Niño–Southern Oscillation: using large ensembles to illuminate time-varying responses and inter-model differences, *Earth Syst. Dynam.*, 14, 413–431, <https://doi.org/10.5194/esd-14-413-2023>, 2023.

Beobide-Arsuaga, G., Bayr, T., Reintges, A. *et al.* Uncertainty of ENSO-amplitude projections in CMIP5 and CMIP6 models. *Clim Dyn* **56**, 3875–3888 (2021). <https://doi.org/10.1007/s00382-021-05673-4>

line 151: see the last comment.

We are unsure what the reviewer was meaning here, and hope that our response to the last comment is clear.

line 159-161: . . . and yet, the exceptions are what confound the canonical cases.

We don't fully understand the point in this comment. This sentence l.159-161 simply indicates the conclusion of the paragraph: the 1-yr estimator is (much) more accurate than its alternative. We have made no changes.

line 201-202: This seems to state the status quo is basically running the show.

We would not speak about “status quo” here, as updating projections with the latest available projection is (really) not a usual practice at the moment. New versions of CMIP are published every 5-7 years, and global-scale attribution / projection analyses have typically been published with a similar frequency. So, even if someone from outside the field may find this result “expected”, it is a really new result, with important implications. The sentence has been revised.

line 222 paragraph: The differences in the regional aspects would be connected to the heat content as well. It might be interesting to consider east coasts and west coasts of continents to quantify if any significant changes emerge.

We would like to mention that there is no statement of how much warming is found in West / East Coasts here. Our finding is really about how to update regular estimates of past / future

warming as new observations become available. Our regional concern is mainly a signal to noise concern. The case under scrutiny is France (i.e., a West coast), but we do not expect key results on the stability of the annual updates to be sensitive to the specific location. That being said, we agree that producing a detailed analysis of how oceanic heat content may affect warming pattern over continents, including east / west coasts, might deliver very useful insights. But this is beyond the scope of our paper, which does not assess spatial warming patterns at all.

line 228-229: How does an estimate for a new ENSO influence the next year's projection?

The influence of particular ENSO event was discussed previously in the manuscript, eg, *“the 2100 SSP2-4.5 warming best-estimate shifted by -0.8°C between [1998 and 2014], followed by a $+0.2^{\circ}\text{C}$ revision induced by the strong El Nino episode in 2015-16”* It was also discussed with respect to the warming to date (although ENSO was not explicitly mentioned):

“the 1-yr estimate has increased by 0.05°C between 2022 and 2023, due to the very high observed GST in 2023”.

We believe this two examples provide a clear illustration of how much particularly hot years (to which ENSO is a key contributor) can influence the estimates.

However, our study does not consider using a *prediction* of an upcoming ENSO event into observationally constrained projections – we only rely on available observations.

Are there biases in the CMIP6 that limit the variances? I.e., are the models not showing enough variance for an upcoming ENSO?

As our focus is to estimate the forced response (the revised method section provides new insights on this point), we make no use of the models' estimate of the ENSO variability – nor other modes of variability. Models are used to specify a prior for the forced response. The statistical property of the internal variability (IV) is then estimated for the real world only, and derived from observations. As we only consider global mean temperature, there is no specific look at ENSO properties – and we also note persistent uncertainty on how ENSO variability will respond to climate change, as mentioned earlier in this point-by-point response.

line 236-237: Does this mean that only the current year is used, not the next year?

Yes – we use all available observations, and since the “next year” has not been observed, it is not used.

line 262-266: What would happen if ECMWF did the same 1-year projection? Does the data assimilation from ECMWF provide similar estimates?

This comment is related to the response we made to R2's general comment ('synopsis'). Our method does not aim to provide initialized projections, i.e. prediction accounting for a particular state of internal variability (eg, as decadal prediction). Instead, it aims to provide the best possible estimate of the forced response, unconditional on the phase of past / future internal variability.

Because our goal is distinct from decadal prediction (and we put no emphasis on predicting next year's temperature at all) we did not make comparison with initialized projections such as suggested by the referee. Additionally, we would like to point out that the ECMWF model is not used to make long-term climate prediction, but if it were, its simulated global mean warming would not be sensitive to the initial conditions beyond the first 5-10 years. There are

also attempts to combine initialized simulations with constrained projections (e.g., Hegerl et al., 2021, *Frontiers*), but this is beyond the scope of our study.

line 301: I see that there could be more information in the early part of the paper to clearly state it is using the available data to estimate the "climate" year from the assimilated data. We were kindly confused by the "climate" year expression, and understand this as the "mean climate" for a given year, i.e. the mean forced response.

We have added one sentence in the first "Result" section to stress that the scope is the forced response, as opposed to internal variability or raw observations.

Another sentence was added to the first paragraph of the "Constrained projections" section.

line 375-376: I would be more interested in the OHC data for the upper 200m of the ocean. For the surface land temperature, I'm not sure if it would benefit from having a heat content approach. The memory in the heat content would be important, for both land and ocean.. We agree with the reviewer that applying a similar method to heat content data would be very attractive, to better assess past and future changes in OHC. To the best of our knowledge, only limited work has been done on this topic so far (see e.g., IPCC WGI AR6, Chapter 9, Fox-Kemper et al., 2021). However, this is a very different question from the one addressed in our study. Our aim here is not to broaden the field of application of observational constraints (many articles do this, and looking at the OHC would be very relevant), but only to assess the interest and relevance of regular updates, on the basis of the most widely used indicator (GST).

Figure 1: "constraint" in the figure is misspelled for the pseudo-observation. Label for "Current year climate estimation" is not really climate, rather it is the annual mean. I think it is better to separate "climate" as a multi-year average as compared to the "annual mean". This doesn't impact the message from my perspective.

The typo is corrected. "climate estimation" was referring to the estimation of the forced response. The word "climate" has been removed from the figure.

Line 460-462: I don't understand why the 10-yr constrained data are biased low. Shouldn't the 10y mean be centered rather than shown at the end of the decadal period?

This is a key point in our study. The usual practice, eg from IPCC is to report the warming over the last 10yr, while we stand at the very end of that observed period. So, basically 10-yr estimates are not centered because there is no way to provide a centered 10-yr average for today... The usual 10-yr estimate is biased for this reason, and this is the reason why we are looking for alternatives. See also Trewin et al. (2022).

Trewin, B. (2022). Assessing internal variability of global mean surface temperature from observational data and implications for reaching key thresholds. *Journal of Geophysical Research: Atmospheres*, 127, e2022JD036747.

<https://doi-org.insu.bib.cnrs.fr/10.1029/2022JD036747>

Figure 3 caption: In Fig 3, I'm struggling to understand why the global and regional figures show the oscillation starting in the mid-70s. This seems to show a lag response that starts ringing with data constraints.

This oscillation is due to low-frequency internal variability of global temperature in the pseudo-observations. The KCC method interprets this as a possible reassessment of the global warming trend (i.e. a reassessment of the system's climate sensitivity). The influence of low-frequency variability on the estimate of climate sensitivity is particularly significant around the year 2000, as this is the time when the forced response starts to emerge, due to the growing magnitude of the forcing and response (i.e. observations start to provide information about climate sensitivity at that time).

The attached figure R1 (below) shows the upper left panel of Fig 3(a), with the corresponding yearly pseudo-observations. Most of the pseudo-observations are above the true forced response during the period 1985-2000, indicative of a positive anomaly in the internal variability, which induces an increase in the assessed (posterior) forced response. On the other hand, during the period 2000-2020, pseudo-observations are mainly below the true forced response, indicative of a negative anomaly in the internal variability, which induces a decrease in the assessed (posterior) forced response. A parallel can be drawn with the "plateau" observed in global mean temperature between the years 1995 and 2015, inducing during the same period a decrease in the estimate of the projected warming (Fig3-b).

Figure R1: Same as Fig3-a with the pseudo-observations (dots, annual global temperature of CNRM-CM6-1 r4i1p1f2) used to constrain, and the global true forced response of CNRM-CM6-1 (dashed grey line, estimated from all available members of the CNRM-CM6-1 model).

I'm puzzled by the jump in France at 1995 and same with the Global at 1975. What is setting these shifts?

We thank Reviewer 2 for this accurate question, which lead us to carefully check the data underpinning the figure, and identify an error. The bottom left panel (c) of the Fig3 was wrong (it showed the regional temperature of the model ACCESS-ESM1-5 r1i1p1f1, instead of CNRM-CM6-1 r4i1p1f2). The figure has been corrected in the revised manuscript. In this new version, there is no more time lag in the jump in France compared to the Global figures. The jump highlighted by the reviewer is still visible in Fig S5 for the ACCESS-ESM1-5 model, quite consistent with GST results (Fig S4, same model), and is presumably related to some particular feature of internal variability in that particular simulation.

line 475: perfect is misspelled.

Corrected

Reviewer #3 (Remarks to the Author):

Review of Ribes et al., Towards annual updating of forced warming to date and constrained climate projections, manuscript number NCOMMS-24-32900.

The submitted paper uses a “perfect model” to argue for the value of providing regular updates to projections of the rise in Global mean Surface Temperature (GST) that are a highlight of Physical Science Basis components of IPCC reports. They make use of output from 8 climate models in the analysis, and they also assess the multi-model mean of 1-yr and 10-yr estimator of the rise in GST.

We thank the reviewer for this fair summary of our work.

Simply, I fail to see the value of the content of this manuscript, especially for a high-profile journal such as Nature Communications.

The question dealt with in this paper, ie, regular updates of key climate change indicators, is in our view a key issue at the moment. It is important in the run-up to AR7, and also in connection with / to inform international climate negotiations. It is also important to update reference trajectories used to inform adaptation strategies. France, for instance, has such a reference trajectory, and its updates are not synchronized with IPCC. This is a new subject, little assessed until now. We believe it is important to specifically assess the pros and cons of this type of calculation, which is the main objective of this study. The regular updating of climate projections in line with the latest observations also seems to us to be a key issue.

Critical issues with climate models such as the so-called “hot model bias” (e.g., Hausfather, Z., Marvel, K., Schmidt, G. A., Nielsen-Gammon, J. W. and Zelinka, M. (2022) 'Climate 1022 simulations: recognize the 'hot model' problem', Nature, 605(7908), pp. 26-29 and references therein) are not addressed. I find the text on lines 143 to 161 to be confusing at best, disingenuous at worse. It is a long-standing problem in the community that the CMIP6 models tend to “run hot”, most likely due to the tendence of the real-world to be in a more La Nina like state over time than is simulated by these models (e.g., Weaver, Clark, et al. "Comparison of proxy-Shortwave Cloud albedo from SBUV observations with CMIP6 models." Journal of Climate 37.11 (2024): 3093-3116 and references therein).

The reviewer raises an important issue concerning models which warm unrealistically strongly or weakly, which was perhaps not clearly enough communicated in our original manuscript. Rather than relying directly on climate model output, which may be biased as the reviewer notes, a key benefit of the method used in our manuscript is that it uses

observed warming to constrain projections of future warming. As noted in the manuscript, results from this and other observational-constraint approaches underpinned global warming projections included in the IPCC AR6, in contrast to earlier IPCC reports which used unconstrained model projections. It is the case that we use climate model output as a prior for our projections, but our imperfect model test results demonstrate that even if some of the climate models we use as priors warm much more than the real world, this does not unduly bias projections of 2100 warming made in the present day.

The “hot models” problem was also discussed in AR6 Chapter 4 (Lee et al., 2021), to some point. As a response to this problem, the AR6 used projections for the 21st century constrained by observations. The methods used, including the KCC method used in our study, were described as capable of correcting the hot models problem. Several previous studies support this, including :

Ribes et al, 2021, Science Advances,
Tokarska et al., 2020, Science Advances,
Brunner et al., 2020, ESD,
Liang et al., 2020, GRL.

The new paragraph on the evaluation of the KCC method provides some of these lines of evidence, in order to show that our statistical method is capable of correctly estimating past and future warming, despite the “hot models” problem. The evaluation of the method even shows that it is capable of correctly predicting the response of models relatively far away from the ensemble mean, including “warm” models (even though there are relatively few of them in the ensemble) and “cold” models (also few). The new comments made on figures S4 and S5 in SI also address this point.

There is concerted effort in the climate community to understand the anomalously high GST in 2023 in papers such as:

Esper, J., Torbenson, M. and Büntgen, U. (2024) '2023 summer warmth unparalleled over the past 2,000 969 years', Nature, 631(8019), pp. 94-97

Schmidt, G. 2024. Climate models can't explain 2023's huge heat anomaly — we could be in uncharted territory. Nature.

Yuan, T., Song, H., Oreopoulos, L., Wood, R., Bian, H., Breen, K., Chin, M., Yu, H., Barahona, D., Meyer, K. 1276 and Platnick, S. (2024) 'Abrupt reduction in shipping emission as an inadvertent geoengineering 1277 termination shock produces substantial radiative warming', Communications Earth & Environment, 5(1), 1278 pp. 281.

none of which are cited. Simply, this paper falls very short of the standard I envision for Nature Communications, and I honestly feel the manuscript will be hard-pressed to review well in a lower profile journal.

The detailed analysis of the strong global mean temperature anomalies observed in 2023 and 2024 is not the main topic of this study (considering that this study would have been relevant even in the absence of a warm record in recent years). Nevertheless, the case of 2023 was already mentioned in the previous version of our manuscript: it is a very recent case, illustrating that a particular observed year can raise important questions about the long-term warming trajectory (including its pace and future evolution), feedbacks, etc. To

reinforce this point, we have followed the reviewer's recommendation and cited several additional papers dealing with the warm anomaly of the recent 1-2 years, including papers that discuss its forced vs internal origin (e.g., references 17–20, ref 45). Furthermore, an important result of our study is to show that, despite a particularly strong warm anomaly in 2023, the revision of constrained projections by including the 2023 observation remains relatively modest. We believe that this result provides further support for the regular updating of observationally constrained climate projections.

Response to reviewers

Title: Towards annual updating of forced warming to date and constrained climate projections

Authors: Aurélien Ribes, Octave Tessiot, Piers Forster, Nathan Gillett, Valérie Masson-Delmotte, Joeri Rogelj, Robert Vautard, Tristram Walsh

We are grateful for the constructive and helpful comments sent by two anonymous referees during this second round of review.

We provide a point-by-point response below (reviewers' comments are in black, our responses in blue), as well as a new version of the manuscript.

REVIEWERS' COMMENTS

Reviewer #1 (Remarks to the Author):

The authors have adequately responded to my comments and I support publication. The paper is now self-contained and I feel like I have a sufficiently complete understanding of what was done to interpret the results.

I do not believe I need to see this again, although I have a couple of suggestions to refine the methods.

We thank the reviewer for this positive appreciation of our revised manuscript.

For the future it would be interesting to see if things change with more sophisticated accounting for structural dataset errors, the effect of structural biases in model sampling, or scenario error.

We are grateful to the reviewer for this stimulating suggestion, and do agree that testing the method against refined uncertainty estimates or adding structural error components will be of high interest. Beyond the updating question discussed in the present manuscript, these are important questions to deal with in order to provide a comprehensive estimation of past and future forced changes. Clearly of interest while preparing the IPCC AR7.

MINOR METHODS COMMENTS

- L351: Methods equation. You use "y" for the observation (forced+ICV) but then " Σ_y " to refer to the covariance *only in the ICV part*. Suggest changing to something like " $\Sigma_{y_{\text{ICV}}}$ " for clarity.

We feel more comfortable keeping with Σ_y for two reasons:

- this y uncertainty doesn't include only ICV, but also measurement uncertainty. In fact our Σ_y is $\Sigma_{y_{\text{ICV}}} + \Sigma_{y_{\text{meas}}}$.
- this notation Σ_y is consistent with the reference publication for the method (Ribes et al., 2021).

The uncertainty on the forced component is not accounted for here because y is considered as an observation of the forced response that is subject to some noise (internal variability + measurement uncertainty). A sentence has been added to make this clearer.

- L360: "therefore including projections – the entire vector x ". If I'm reading correctly I think you mean "of the entire vector x ". Please check.

The sentence has been slightly revised:

As a result, this method provides observationally constrained estimates of past, present and future forced warming (i.e., the entire vector x), including projections.

- L335-338 is an example of some of the clunkier writing in the new Methods text, with three versions of the word "estimate" in it. I think readers would appreciate it if the authors skimmed once again through the Methods to bring the language and its flow up to the standards of the rest of the paper.

We have carefully reviewed and corrected the language in the Method section – see the track change file for more details about this. We think the revised version now meets a high standard.

Reviewer #2 (Remarks to the Author):

After reading the R2 version of the manuscript, I am satisfied with the revision. This paper has been revised is very I have also read through the two additional reviews. From my perspective, I am ready to accept the current version of the manuscript for publication. If needed, I would be available to review a third version of the manuscript.

We thank the reviewer for this positive appreciation of our revised manuscript.